# Teaching Bioinspired Design for Assistive Technologies Using Additive Manufacturing: A Collaborative Experience

**DOI:** 10.3390/biomimetics10060391

**Published:** 2025-06-11

**Authors:** Maria Elizete Kunkel, Alexander Sauer, Carlos Isaacs, Thabata Alcântara Ferreira Ganga, Leonardo Henrique Fazan, Eduardo Keller Rorato

**Affiliations:** 13D Orthotics and Prosthetics Laboratory, Science and Technology Institute, Federal University of São Paulo UNIFESP, São José dos Campos 12247014, Brazil; elizete.kunkel@unifesp.br (M.E.K.); thabata.ganga@unifesp.br (T.A.F.G.); leonardohfazan@gmail.com (L.H.F.); ek.rorato@unifesp.br (E.K.R.); 2Institute for Bionics, Westphalian University of Applied Sciences, 46397 Bocholt, Germany; 3Department of Chemical Engineering, Universidad de La Frontera UFRO, Temuco 4811230, Chile; carlos.isaacs@ufrontera.cl

**Keywords:** bioinspired design, additive manufacturing, prosthetics and orthotics, technology for disability, engineering education, Bionik, 3D printing, assistive technology

## Abstract

Integrating bioinspired design and additive manufacturing into engineering education fosters innovation to meet the growing demand for accessible, personalized assistive technologies. This paper presents the outcomes of an international course, “3D Prosthetics and Orthotics”, offered to undergraduate students in the Biomimetic program at Westfälische Hochschule (Germany), in collaboration with the 3D Orthotics and Prosthetics Laboratory at the Federal University of São Paulo—UNIFESP (Brazil). The course combined theoretical and hands-on modules covering digital modeling (CAD), simulation (CAE), and fabrication (CAM), enabling students to develop bioinspired assistive devices through a Project-based learning approach. Working in interdisciplinary teams, students addressed real-world rehabilitation challenges by translating biological mechanisms into engineered solutions using additive manufacturing. Resulting prototypes included a hand prosthesis based on the Fin Ray effect, a modular finger prosthesis inspired by tendon–muscle antagonism, and a cervical orthosis designed based on stingray morphology. Each device was digitally modeled, mechanically analyzed, and physically fabricated using open-source and low-cost methods. This initiative illustrates how biomimetic mechanisms and design can be integrated into education to generate functional outcomes and socially impactful health technologies. Grounded in the Mao3D open-source methodology, this experience demonstrates the value of combining nature-inspired principles, digital fabrication, Design Thinking, and international collaboration to advance inclusive, low-cost innovations in assistive technology.

## 1. Introduction

The integration of bioinspired design and additive manufacturing into engineering education represents a promising path to address the current challenges in the development of accessible and personalized assistive technologies [1,2,3]. Bioinspired design is an innovative approach that seeks to solve human challenges by observing and emulating strategies found in nature [4]. This method draws from millions of years of natural evolution to inform engineering solutions that are innovative, efficient, and sustainable [5,6,7]. For example, the burrs that tenaciously clung to his dog’s fur inspired the Swiss engineer Georges de Mestral to invent Velcro, a fastening system of tiny hooks and loops that was remarkably innovative for its time [8].

Additive manufacturing (AM), also known as three-dimensional (3D) printing, refers to the process of fabricating objects layer by layer from digital models, enabling the creation of complex geometries with high precision and minimal material waste [9,10,11]. In contrast to traditional subtractive methods, AM provides greater flexibility in design and rapid prototyping, which is particularly advantageous in healthcare [12,13,14,15]. It has proven especially valuable in the development of personalized assistive devices, such as prostheses and orthoses, tailored to the unique anatomical and functional needs of users [16,17,18]. This includes bioinspired workflows for lower limb prosthetic design, such as the development of monocoque transtibial prostheses using additive manufacturing [19].

Upper limb orthoses, for instance, have been produced with custom geometry to enhance user comfort and biomechanical performance [20,21,22,23]. Similarly, additive manufacturing has enabled the creation of a wide range of low-cost prosthetic hands, particularly for upper limb differences, offering functional, esthetic, and psychosocial benefits [24,25,26,27,28]. Facial and breast prostheses manufactured via AM have also demonstrated promising clinical and psychosocial outcomes [29,30,31]. This design freedom and potential for personalization make additive manufacturing a key enabler of inclusive, cost-effective solutions in rehabilitation and assistive technologies [32,33,34,35].

By combining these three approaches, nature-inspired design, digital fabrication, and user-centered customization, it becomes possible to create assistive solutions that are more accessible, comfortable, and functional, ultimately enhancing quality of life and promoting social inclusion [36,37,38].

As the global population ages and the prevalence of chronic conditions and physical disabilities increases, there is a growing demand for cost-effective, personalized solutions that can be tailored to the individual needs of each user [39]. Bioinspired design offers a creative and ergonomic framework for addressing these challenges [40,41,42], and when combined with the flexibility of additive manufacturing, it enables rapid prototyping and customization, even in resource-constrained settings [43]. Embedding these technologies into engineering curricula fosters not only technical skills, but also ethical reflection on interdisciplinary collaboration, preparing future professionals to create inclusive and socially impactful innovations [44,45,46]. Recent educational strategies have emphasized creativity in design for AM, with interventions that promote divergent thinking and innovation among students [47]. Egan [48] also highlights the importance of aligning AM design practices with sustainability and emerging technologies to prepare engineers for future challenges.

Despite the growing global relevance of this field, there is a noticeable shortage of professionals worldwide who possess the interdisciplinary skills necessary to operate at the intersection of additive manufacturing, assistive technology, and biomimetics [49,50,51]. This gap poses a challenge to the advancement and large-scale implementation of innovative, personalized solutions in healthcare and rehabilitation. Furthermore, formal education programs that effectively integrate these domains in a coherent, applied, and user-centered manner remain scarce across the globe, particularly those focused on affordable and socially impactful applications [52,53,54].

Project-based learning (PBL) is an innovative student-centered education strategy that involves interdisciplinary teaching and learning activities [54,55,56,57]. It is a technology-based approach that promotes active and collaborative learning, in which students are guided by experienced instructors and acquire deeper knowledge by responding to real-world questions, problems, and challenges [58,59,60], particularly within interdisciplinary fields such as biomimetics, where nature-inspired solutions are applied to complex engineering and societal needs [61]. Co-design approaches involving end users, as proposed by Withana, enrich this learning model by fostering personalization, engagement, and real-world applicability [38].

The Mao3D program is a technology-based, open-source social innovation initiative developed over the past 10 years at the 3D Orthotics and Prosthetics Laboratory of the Federal University of São Paulo (LO&P3D/UNIFESP), a public university in Brazil [62]. Its primary goal is to democratize access to assistive technologies by promoting the use of low-cost 3D printing for the development of personalized orthoses and prostheses. Beyond its technical contributions, Mao3D plays a key role in capacity-building by training students, health professionals, and makers to collaboratively design and fabricate assistive devices tailored to users’ needs. The program fosters interdisciplinary learning and empowers participants with skills in digital modeling, bioinspired design, and additive manufacturing, preparing them to work ethically and effectively in the field of rehabilitation and inclusive technology.

This article presents the outcomes of an international teaching initiative centered on the course ‘3D Prosthetics and Orthotics’. The course was offered to undergraduate students enrolled in the Bachelor’s degree of Biomimetics at the Institut für Bionik, Westfälische Hochschule (Germany), in collaboration with the LO&P3D/UNIFESP, focusing on applied bioinspired design and structural optimization principles rooted in the German tradition of Bionik [63]. Practical activities included the modeling and customization of open-source prosthetic devices, such as the Unlimbited Arm [64], developed and maintained by the global e-NABLE community [65].

## 2. Materials and Methods

This study employed a qualitative, exploratory methodology structured around an international, interdisciplinary teaching experience, aligned with the principles of research-based and practice-oriented pedagogy. The choice of a qualitative approach was guided by the need to understand processes of knowledge construction and collaborative innovation within real-world learning environments. Methodologically, this initiative can be characterized as a research-action format within the field of engineering education [45]. Implemented as an intensive two-week module in Bocholt (Germany), the course was part of a collaborative initiative between the Federal University of São Paulo (UNIFESP) and Westfälische Hochschule (WHS). The instructional design was based on PBL and emphasized iterative prototyping, user-centered approaches, and digital workflows involving Computer-Aided Design (CAD), Computer-Aided Engineering (CAE), and Computer-Aided Manufacturing (CAM).

### 2.1. Course Overview: Educational Objectives and Learning Approach

A total of 13 undergraduate students enrolled in the Bachelor’s program in Biomimetics participated in the elective course “3D Prosthetics and Orthotics”, which was held in July 2023 at the WHS campus in Bocholt, Germany. The course was developed for undergraduate students who had completed CAD and CAE courses previously. The “3D Prosthetics and Orthotics” course consists of lectures, laboratory training, and real-world projects. Since 2022, the course content has been updated annually according to teaching team experience and students’ feedback. Student evaluation is based on the grades obtained individually on the midterm test and a final examination, and on the grade achieved collectively by the semester-long project team.

The 60 h course was structured in a bilingual format, with approximately half of the content delivered through theoretical modules in German and the other half through practical modules in English. Its main objective was to equip students with the skills to apply bioinspired design and additive manufacturing in the development of accessible and personalized assistive technologies. The program emphasized interdisciplinary collaboration, empathy-driven design, and hands-on experimentation in real-world contexts.

### 2.2. Theoretical and Practical Modules

The course combined theoretical instruction with practical laboratory experience (Figure 1). The theoretical component introduced students to key concepts in biomimetics, such as biological analogies applicable to assistive technology, functional anatomy and biomechanics, digital modeling for rehabilitation, and the fundamentals of materials and extrusion-based 3D printing. This part of the course presented the possibilities of manufacturing prostheses and orthotics for various parts of the body using the advanced manufacturing process. These sessions included case studies, semantic analysis panels, and discussions on the ethical and social impact of assistive technology in Brazil and Germany. References supporting the theoretical foundations of biomimetic education were used to frame the biological inspiration process [1,64].

During the practical modules, students first engaged in hands-on training with digital tools commonly used in assistive technology design. This included 3D modeling software for customizing prosthetic components, slicing software for preparing digital models for printing, and simulation tools to assess mechanical performance. These software-based activities provided a foundational understanding of the digital workflow involved in additive manufacturing processes. Building on this experience, students then applied protocols developed by the Mao3D program [62] to fabricate an open-source mechanical upper limb prosthesis, the Unlimbited Arm v2 [64], as an introductory project. This task combined their software skills with hands-on assembly, using the fused filament fabrication (FFF) process, also known as fused deposition modeling (FDM, a registered trademark of Stratasys). The integration of digital and physical practices offered a comprehensive perspective on the development of bioinspired assistive technologies through 3D printing.

In the subsequent sessions, in order to develop an effective PBL strategy, five steps, based on Stern et al. [53], were defined and adapted for each one of four student groups: (1) a driving question (based on real-world dilemmas) was required; (2) cooperative learning by forming heterogeneous group collaboration was advisable; (3) investigation should be performed and the students should offer several solutions; (4) an experimental part of the project should be based on the CAD, CAE, and CAM tools and biomimetics principles; and (5) the final product should be presented. Each student team engaged in the full development cycle of an original assistive technology device tailored to a specific fictional user scenario including the ideation, modeling, simulation, and fabrication of personalized orthoses or prostheses.

To support this process, students utilized a comprehensive suite of digital tools for Computer-Aided Design (CAD), Computer-Aided Engineering (CAE), and Computer-Aided Manufacturing (CAM), enabling the efficient design, simulation, and fabrication of their projects. The tools and 3D printers available in the WHS laboratory that supported the fabrication process are summarized in Table 1 and Table 2. These resources were integrated into the practical workflow in accordance with the specific requirements of each project. Most components were printed using Polylactic Acid (PLA), selected for its ease of use and wide availability in the laboratory. Additionally, Thermoplastic Polyurethane (TPU) was employed for components requiring enhanced flexibility, providing valuable functional characteristics tailored to the needs of specific student projects.

### 2.3. Project Development Process

In the final two-hour segment of each course day, biomimetics students worked in teams on project-based tasks, applying previously acquired concepts and tools to develop original assistive devices grounded in biomimetic principles and tailored to real-life case scenarios. Building upon the five-step PBL framework previously established, this phase of the course introduced an expanded eight-step pedagogical model structured through a Design Thinking lens (Table 3). This enriched approach retained the original PBL elements, such as problem definition, collaborative investigation, and hands-on experimentation, while incorporating additional stages focused on empathy, ideation, iterative prototyping, and user-centered evaluation. Within this structure, student prototypes evolved through continuous feedback and refinement, fostering deeper engagement with real-world problem-solving and design innovation in assistive technologies.

Although the projects were based on fictional user scenarios, students were encouraged to validate their solutions through usability tests and scenario-based simulations. However, due to the academic nature of the course, no direct testing with real users was performed. This limitation is acknowledged and suggests an opportunity for the future implementation of pilot studies with actual end users.

Throughout the course, students were mentored by faculty from WHS and UNIFESP, who provided guidance on technical, clinical, and ergonomic aspects of the projects.

### 2.4. Evaluation and Feedback

Projects were evaluated based on multiple criteria: innovation, usability, functionality, feasibility, and alignment with user needs. Student performance was also assessed in terms of engagement, teamwork, and problem-solving skills. Feedback was gathered qualitatively through final presentations, peer reviews, and guided group reflections. The qualitative data collected from peer review sessions and group discussions were analyzed using a thematic approach, allowing the teaching team to identify patterns in student learning, challenges encountered, and innovative strategies developed throughout the course. The course structure enabled an environment of co-creation and critical reflection, helping students to build a professional identity rooted in socially responsible engineering practices.

The evaluation framework for the “3D Prosthetics and Orthotics” course was structured to integrate progressive skill acquisition with a final assessment focused on the development of a functional assistive device. This methodological approach ensured that students could build upon prior knowledge incrementally while advancing toward the creation of a complex, 3D-printed orthopedic device. The assessment model was designed to assess principles of human-centered design, technical engineering, and scientific validation. The evaluation process was structured around a sequence of progressive deliverables:The empathy map: introduction to human-centered design through analysis of the lived experiences of individuals with physical disabilities;Literature review: identification of validated solutions and design gaps through scientific inquiry;Technical drawings: translation of conceptual ideas into 2D schematics;3D modeling (CAD): spatial reasoning and manufacturability considerations;Mechanical simulation using Finite Element Modeling (FEM): performance analysis under mechanical loads;Initial prototyping: physical 3D printing of the first version;Iterative refinement: design optimizations based on testing and evaluation;Final prototype: functional assistive device integrating all prior stages.

The empathy map emphasized the necessity of empathetic design, reinforcing that the development of assistive devices extends beyond technical proficiency and necessitates a deep understanding of the end user’s lived experiences. Beyond the technical deliverables, students were also required to demonstrate scientific communication skills. Two final assessments were conducted in English:Pitch presentation (5 min): communicating innovation and impact succinctly to academic and professional audiences;Scientific report: documentation of the design process, validation methods, and critical reflections following academic standards.

## 3. Results

This section presents the outcomes of the “3D Prosthetics and Orthotics” course based on the same structure outlined in the methodology, allowing a direct comparison between pedagogical intentions and achieved results.

### 3.1. Course Overview: Educational Outcomes

The course brought together 13 undergraduate students, who were divided into four interdisciplinary teams. On the first day, students were introduced to the fictional user scenarios that would guide their project development. In small groups, they began outlining initial ideas for personalized assistive technologies, guided by empathy-driven design principles. Despite the challenge posed by the bilingual structure, where theoretical content was delivered in German and practical content in English, the experience facilitated a rich cultural and academic exchange. For the Brazilian mentors, the language barrier initially posed a difficulty in following the theoretical sessions, while German students reported having to adapt to receiving practical guidance in English. However, this dynamic fostered a high level of collaboration and engagement from both sides. The educational objectives of fostering intercultural communication, applied biomimetic design, and hands-on innovation were successfully addressed through collaborative activities and reflective assessments.

### 3.2. Theoretical and Practical Integration

The daily routine was structured to support deep immersion: each day began at 9 a.m. with theoretical sessions conducted in German, covering core topics such as biomimetics, functional anatomy, biomechanics, and the digital fabrication of devices for rehabilitation. After lunch, students participated in hands-on practical workshops facilitated by the Brazilian mentors. During these sessions, students learned to operate desktop 3D printers, fabricate and assemble the open-source prosthesis Unlimbited Arm v2 from scratch, and design custom orthoses using various CAD tools.

Although students already had experience using parametric engineering CAD software (e.g., Simens NX, Fusion 360, TinkerCAD), most had never worked with organic modeling tools such as Blender and Meshmixer. These tools were introduced in the workshops, where students applied them directly to their own projects, gaining familiarity through practice. The integration of theory and practice was seamless, with practical tasks reinforcing theoretical knowledge and providing opportunities to iterate their project designs. The bilingual dynamic is exemplified in Figure 2, which complements the integration between theoretical and practical modules. Part A shows a theoretical lecture delivered in German, where foundational concepts in assistive technology devices were discussed. Part B depicts a 3D scanning workshop conducted in English, where students learned how to use the scanning tool in practice. This bilingual learning environment, while initially challenging, promoted flexibility and fostered peer-to-peer learning across linguistic and cultural boundaries.

As part of the practical modules, students were introduced to the e-NABLE community, a global maker network focused on open-source assistive devices. They explored its online repository and identified the prosthesis Unlimbited Arm v2 as the most widely adopted and validated prosthetic design. This model is preconfigured in OpenSCAD and requires only basic anthropometric measurements to generate customized 3D files. After understanding the generation process, students engaged in slicing the models and discussed key 3D printing parameters, such as the infill density, layer height, support structures, and material selection. Once the parameters were defined, the prosthesis was printed using the FFF process. During the prosthesis assembly workshop, students participated in all major steps: thermoforming components for a better fit, inserting joint pins, setting up the actuation mechanism using fishing line, and performing surface finishing and covering. The activity concluded with a fully functional printed prosthesis, integrating digital design knowledge with hands-on fabrication and assembly skills (Figure 3).

### 3.3. Project Development by Students

Four original assistive devices were developed by student teams, each aligned with fictional user scenarios provided at the beginning of the course. The development followed the Design Thinking methodology described in the methods section. Students progressed through empathy mapping, ideation, digital modeling, simulation, prototyping, and testing. Table 4 summarizes the projects, including the device type, team size, and biomimetic inspiration behind each design.

These student-led projects exemplify the educational impact of combining biomimetics, additive manufacturing, and open-source design. By engaging with realistic challenges and using digital fabrication tools, students created functional and socially relevant solutions. The process also cultivated critical thinking, collaboration, and awareness of inclusive design principles. In future iterations of the course, involving real users in selected stages (such as testing or co-design workshops) could enhance the translational potential of the outcomes and foster deeper empathy.

#### 3.3.1. Finger Prosthesis for Musician

This team designed a lightweight, body-powered prosthetic finger for a user with partial hand amputation, following the eight-step PBL model. Inspired by the biomechanics of bird phalanges, the project emphasized simplicity, functional articulation, and affordability.

Problem Definition: The user scenario emphasized challenges in performing tasks requiring opposition and grasp, such as picking up small objects. The goal was to restore basic hand functions using a mechanically simple, non-electronic solution;Biological Research: Avian anatomy—particularly bird phalanges known for lightness and efficient force transmission—inspired a segmented, angled finger design that could replicate essential joint motion;Ideation: Early concepts were sketched using semantic panels to explore mechanical function and comfort. Elastic bands and strings were proposed for passive actuation (Figure 4);

4.Decision-making: A decision matrix guided the selection of PLA filament for rigidity and dental rubber bands for return force, balancing cost, printability, and functional response;5.Modeling and Simulation: The design was modeled in Fusion 360. FEM stress analysis was applied to validate mechanical performance, focusing on critical joints (Figure 5);

6.Fabrication: The prototype was printed using a Prusa Mini printer. Articulated joints were manually assembled with embedded elastic elements (Figure 6).

7.Testing and Evaluation: The prosthesis was evaluated in simulated tasks (e.g., gripping a pen, holding a spoon). Iterations improved the return force and fit based on performance feedback;8.Presentation and Reflection: In the final presentation, students demonstrated functionality and discussed lessons on alignment, iterative design, and adaptability. The process highlighted the educational value of connecting biomechanics with accessible prototyping tools.

#### 3.3.2. Handy Solutions

This project focused on a modular finger prosthesis with interchangeable tips, targeting both functional performance and psychosocial aspects of partial hand amputation. The design emphasized user-centered customization, low cost, and playful interaction—especially relevant for pediatric applications.

Problem Definition: Finger amputations impact daily function and social inclusion. Commercial solutions are often expensive and lack personalization. The goal was to develop a functional, affordable, and customizable prosthesis for distal and intermediate phalanx loss;Biological Research: While not based on a specific animal model, the design was inspired by the tendon–muscle antagonism found in human anatomy. Fishing lines simulated flexor tendons, while rubber bands enabled extension, reproducing basic biomechanical function;Ideation: Initial sketches proposed a tendon-driven, modular prosthesis with simplified assembly and plug-in fingertip modules (Figure 7). Playful extensions, like a stylus or flashlight, supported engagement and personalization;

4.Decision-making: PLA was selected for its ease of printing, and brass rods were later added to reinforce joints after FEM analysis revealed shear stress issues. Simplicity and compatibility with desktop 3D printers were key decision factors;5.Modeling and Simulation: Design iterations were modeled in Siemens NX. FEM simulations under a 10 N fingertip load identified critical stress regions in joint bolts, guiding adjustments to materials and geometry (Figure 8);

6.Fabrication: Three prototypes were printed using Prusa Mini and Anycubic Chiron printers (Figure 9). Progressive iterations addressed tendon path scaling, joint dimensions, and printability. The final version integrated brass joints and improved articulation;

7.Testing and Evaluation: Prototypes were tested for movement, fit, and structural integrity. Key issues included hyperextension and poor phalanx synchronization. Redesigns refined the rubber band positioning, cap alignment, and joint performance;8.Presentation and Reflection: The group presented their design journey, engineering decisions, and testing outcomes. Reflections addressed challenges in joint design and team coordination. Future directions include enhancing grip precision and refining modular attachments for broader user engagement.

#### 3.3.3. Cervical Brace

This team developed a pediatric cervical orthosis aimed at improving comfort, usability, and anatomical adaptability. The design was inspired by stingray fins, which distribute pressure evenly while maintaining flexibility.

Problem Definition: Conventional cervical braces often cause discomfort and are difficult to apply. The project aimed to create a brace that reduces pressure points, allows better airflow, and is easier to wear independently—particularly for children;Biological Research: Inspired by stingrays, the team adopted smooth, broad contact surfaces and curved forms to ensure even pressure distribution and flexibility, promoting both stability and comfort;Ideation: Initial sketches explored structural concepts emphasizing ventilation, ergonomic fit, and lightweight construction (Figure 10). Frontal and lateral views guided proportion decisions before digital modeling;

4.Decision-making: PLA was selected due to its printability and lab availability, prioritizing feasibility over material flexibility. Although alternatives like ABS or TPU offer improved mechanical properties, they were less compatible with the available open printers;5.Modeling and Simulation: The design process began with geometric primitives and evolved into an anatomically adapted model. A digital human mesh was created in MakeHuman and refined in Blender using sculpt tools. FEM analysis identified low-stress regions and enabled material reduction for improved ventilation without compromising strength (Figure 11);

6.Fabrication: The final model was printed in PLA using a Bambu Lab X1 Carbon with 14% infill and tree supports, completed in approximately 9 h with no post-processing required (Figure 12).

7.Testing and Evaluation: The orthosis was evaluated on both a digital model and a user with matching anthropometry (Figure 13). Assessments focused on fit, comfort, and restriction of cervical motion. Identified issues included excess pressure at the chin and limited ventilation;

8.Presentation and Reflection: The final presentation highlighted the brace’s ergonomic improvements and ease of application. Peer feedback emphasized the functional fit and esthetic quality. The team reflected on challenges such as anatomical variation and proposed future enhancements using thermoformable materials and adjustable closure systems.

#### 3.3.4. Fin Ray-Inspired Hand Prosthesis

This team explored the Fin Ray effect to develop a prosthetic hand with passive adaptability. The design aimed to enable fingers to conform naturally to objects, improving the user’s ability to handle irregular shapes through mechanical compliance.

Problem Definition: Low-cost prosthetic hands often lack grip adaptability, while high-end myoelectric models are expensive and complex. This project aimed to create an affordable, mechanically actuated hand capable of secure and adaptive grasping;Biological Research: The Fin Ray effect, observed in the tail fins of fish such as trout, allows flexible structures to bend toward applied forces. This principle informed the design of adaptive fingers that could conform to objects during grip (Figure 14).

3.Ideation: Initial sketches included a sport-specific prosthesis and embedded tools. The team later shifted toward a modular, general-purpose design incorporating Fin Ray geometry (Figure 15).

4.Decision-making: PLA was used for rigid components and TPU for the flexible fingers, balancing performance and printer compatibility. Several failed prints guided material selection and structural simplifications;5.Modeling and Simulation: CAD models were developed in Siemens NX, evolving through three iterations. The final design included two adaptive fingers and a central support. FEM analysis simulated a 100 N load on connecting pins, guiding reinforcement (Figure 16);

6.Fabrication: Prototypes were printed using Prusa MK4 and Mini printers. While finger modules were successfully fabricated, full assembly was not completed due to design complexity and print failures;7.Testing and Evaluation: Only partial testing was possible. Isolated components revealed issues with finger stability and assembly tolerance. The process underscored the challenges of translating bioinspired geometry into reliable low-cost prototypes;8.Presentation and Reflection: In the final presentation, the team highlighted their design process, analytical work, and lessons learned. Despite the absence of a fully functional prototype, students reflected on the value of iterative modeling, material constraints, and simplifying complex ideas for feasible implementation.

### 3.4. Performance Evaluation

The final presentations were held at the conclusion of the two-week program, during which each team showcased their prototypes and reflected on their development process. Peer evaluation and instructor feedback emphasized the originality of the solutions, the use of biomimetic principles, and the clarity of technical communication. Students reported that the course enhanced their understanding of user-centered design and broadened their perspective on the application of biomimetics to assistive technology. Many highlighted the opportunity to work with international mentors and new software tools as transformative aspects of the course.

The deliverable-based evaluation approach led to improvements in skill acquisition and project outcomes. A key outcome was the impact of user-centered research on design quality. The integration of empathy mapping and the literature review facilitated a transition from assumption-based to evidence-driven design, ensuring deeper alignment with user needs. Many students significantly refined their initial concepts based on these research tools, resulting in final prototypes that exhibited enhanced functionality and ergonomics.

Students demonstrated technical proficiency gains in 3D modeling, FEM analysis, and digital fabrication. Computational simulations optimized structural integrity and material efficiency, leading to more durable and functional designs. The iterative transition from technical drawings to 3D models and physical prototypes reinforced the relationship between design theory and manufacturability. Notably, students who engaged more extensively in iterative optimization produced higher-performing prototypes, highlighting the importance of systematic testing and validation.

The structured prototyping and refinement process also strengthened problem-solving abilities. By analyzing prototype failures and iterating improvements, students applied scientific reasoning to design modifications, ensuring their assistive devices met practical and manufacturability criteria. This iterative methodology underscored the central role of engineering validation in biomedical device development.

Beyond technical execution, final assessments, the pitch presentation and scientific report, were critical to professional development. The pitch required students to distill project innovations into a concise and persuasive narrative, enhancing their ability to communicate technical concepts effectively to diverse audiences. The scientific report formalized their findings, documenting research outcomes, methodological justifications, and engagement with academic discourse. These components ensured that students not only gained expertise in engineering design and prototyping, but also essential scientific communication skills, critical for careers in medical technology development.

The limitations of this iteration included the short course duration, restricted testing with real users, and occasional equipment constraints. Future versions of the course may include pilot tests with target users and more robust usability evaluations.

## 4. Discussion

The integration of bioinspired design and additive manufacturing into engineering education, as demonstrated in this international course, offers a promising pathway to address the growing global demand for accessible, personalized assistive technologies. As highlighted by the WHO [39], access to assistive products remains limited for over one billion people worldwide, with disproportionate impacts in low- and middle-income countries. In Brazil, despite the existence of a universal healthcare system, access to individualized orthoses and prostheses is often centralized, slow, and dependent on scarce specialized services. In contrast, Germany benefits from broader institutional support for assistive technologies, yet still faces challenges related to cost, user-centered customization, and the adoption of innovative fabrication methods.

The results observed with the course “3D Prosthetics and Orthotics”, particularly the development of original assistive prototypes and the students’ ability to iterate designs based on biomimetic principles, confirm the viability and educational value of integrating digital fabrication and nature-inspired thinking into real-world problem solving. In both contexts, open-source methodologies and additive manufacturing offer a disruptive alternative to traditional prosthetic production. As emphasized by Ngo et al. [9], 3D printing allows for the fabrication of complex, customized geometries with reduced waste and cost, facilitating decentralized and user-driven innovation. The open-source Mao3D approach, which grounded the practical modules of this course, exemplifies how grassroots, low-cost solutions can empower communities to design and produce assistive devices tailored to individual needs [62].

The biomimetic approach, central to the Biomimetics curriculum at WHS, provided a creative and functional framework for the students’ projects. Inspired by nature’s principles of adaptability, resilience, and efficiency [1,63], students were encouraged to observe biological systems and translate these insights into technical designs. This aligns with the findings of Vincent et al. [1] and Arena et al. [6], who advocate for biomimetics as a driver of innovative problem-solving in healthcare and rehabilitation contexts. The interdisciplinary and international format of the course further amplified the learning outcomes. Working in cross-cultural teams enriched students’ collaborative abilities, fostered empathy, and introduced diverse problem-solving strategies, skills that are increasingly essential in global engineering practice.

Importantly, the development of prototypes inspired by biological models was not treated as an abstract design exercise, but as a scientific and technical challenge grounded in real functional requirements. Each team was required to apply biomimetic reasoning to transform biological inspiration into mechanical performance, using CAD modeling, FEM simulations, and material selection strategies to validate design choices.

The observed outcomes in student engagement, creativity, and technical skills support the view that hands-on, interdisciplinary learning environments enhance engineering education. Similarly to the approach discussed by Golecki et al. [37], students in this course applied Design Thinking methodologies to real-world challenges, progressing from empathy mapping to prototype iteration. They developed not only functional assistive devices but also a sense of ethical responsibility and empathy, key competencies for engineers working at the interface of technology and health.

Nevertheless, some limitations were identified. The two-week duration of the course limited the depth of mechanical testing and constrained opportunities for iterative refinement based on user feedback. Additionally, while additive manufacturing enabled rapid prototyping, some teams encountered difficulties with mechanical tolerances, material properties, and printer calibration. These challenges reflect the broader barriers to implementing 3D-printed prosthetics at scale, as discussed in Zuniga et al. [24], and underscore the need for continued technical training and infrastructure investment.

To address these limitations, future editions of the course could adopt a hybrid format with preparatory online modules and extended project development timelines. Involving end users and healthcare professionals earlier in the process may also enhance the relevance and usability of the designs. Establishing long-term collaborations and incubating student projects, possibly through international partnerships, could further bridge the gap between educational prototypes and clinical implementation.

In this context, the course also contributes to strengthening the field of biomimetics by offering a replicable strategy for training professionals capable of translating biological systems into engineered devices. Moreover, this teaching experience suggests a scalable pedagogical model that can be adapted to a variety of educational settings, including undergraduate programs, continuing education, and technical training in healthcare innovation.

Finally, the course model demonstrates strong potential for replication in diverse contexts. As noted by Stern et al. [53] and Ranger & Mantzavinou [44], embedding digital fabrication and biomimetic thinking into engineering curricula not only fosters technical competency, but also promotes social impact and inclusion. The combination of low-cost tools, open knowledge, and intercultural collaboration creates a scalable framework for empowering the next generation of socially engaged engineers.

## 5. Conclusions

This international teaching initiative demonstrated the educational and social value of integrating biomimetic principles, additive manufacturing, and assistive technology into undergraduate engineering education. Through hands-on, project-based learning, students not only acquired technical skills in CAD, CAE, and CAM tools, but also cultivated empathy, creativity, and critical thinking while tackling real-world rehabilitation challenges.

The interdisciplinary and intercultural collaboration between Brazil and Germany enriched the learning process by exposing students to diverse healthcare systems and fostering dialog on the role of low-cost innovation in promoting equity and inclusion. The application of biological models, such as the Fin Ray effect, tendon–muscle antagonism, and stingray-inspired morphology, enabled the development of accessible, customized assistive solutions that balanced functionality with esthetic and ergonomic considerations.

The successful development of four original bioinspired prototypes, combining digital modeling, mechanical simulation, and additive manufacturing, validated the methodological approach and confirmed the course’s impact on applied learning. These outcomes emphasize the value of integrating technical, ethical, and human-centered dimensions into engineering education, while also contributing concrete results to the field of biomimetic design and fabrication.

Key lessons from this experience include the effectiveness of user-centered design, the pedagogical power of iterative prototyping, and the potential of open-source digital fabrication in academic contexts. To further enhance the translational impact of such educational efforts, future iterations of the course should incorporate extended timelines, the early involvement of real users, and structured follow-up mechanisms, such as clinical testing, co-design workshops, and mentorship programs, to support the maturation and validation of promising student innovations.

Moreover, this initiative contributes to the consolidation of biomimetics as a field of applied research by demonstrating its feasibility as a framework for the design and fabrication of functional health technologies. Ultimately, this teaching model offers a replicable and scalable framework that can be adapted to a wide range of educational settings, including undergraduate programs, technical training, and continuing education. By aligning biomimetic design and digital manufacturing with socially responsible values, courses like “3D Prosthetics and Orthotics” contribute to shaping a new generation of engineers capable of creating meaningful, inclusive, and context-sensitive health technologies.

## Figures and Tables

**Figure 1 biomimetics-10-00391-f001:**
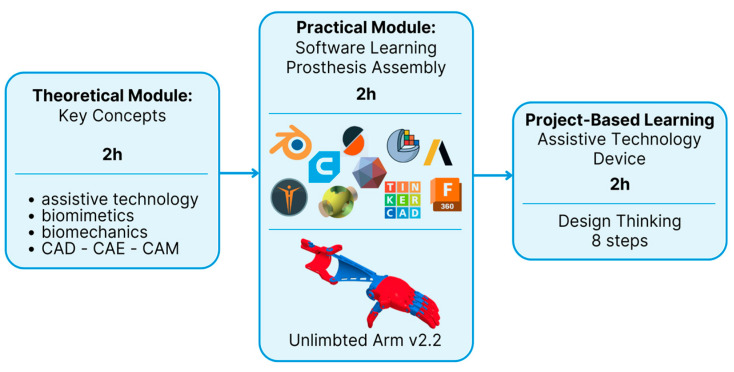
Flowchart of the methodology used in the “3D Prosthetics and Orthotics” course. It combines theoretical and practical approaches with CAD, CAE, and CAM tools, nature-inspired design (biomimetics), 3D printing, and the development of assistive technology projects. The methodology is based on real-world problem solving, with an interdisciplinary and collaborative focus, promoting innovation, sustainability, and social inclusion.

**Figure 2 biomimetics-10-00391-f002:**
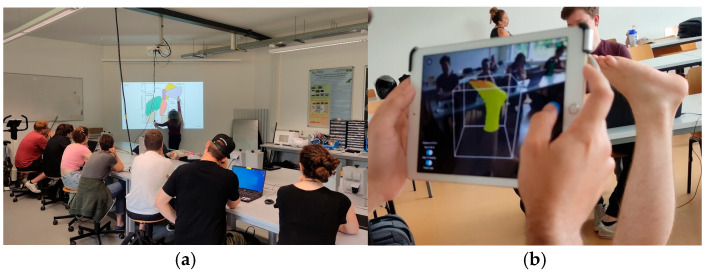
Moments captured from the “3D Prosthetics and Orthotics” course showcasing the bilingual learning environment. (**a**) Example of a theoretical lecture delivered in German. (**b**) Practical 3D scanning workshop taught in English, where students learned how to operate the scanner.

**Figure 3 biomimetics-10-00391-f003:**
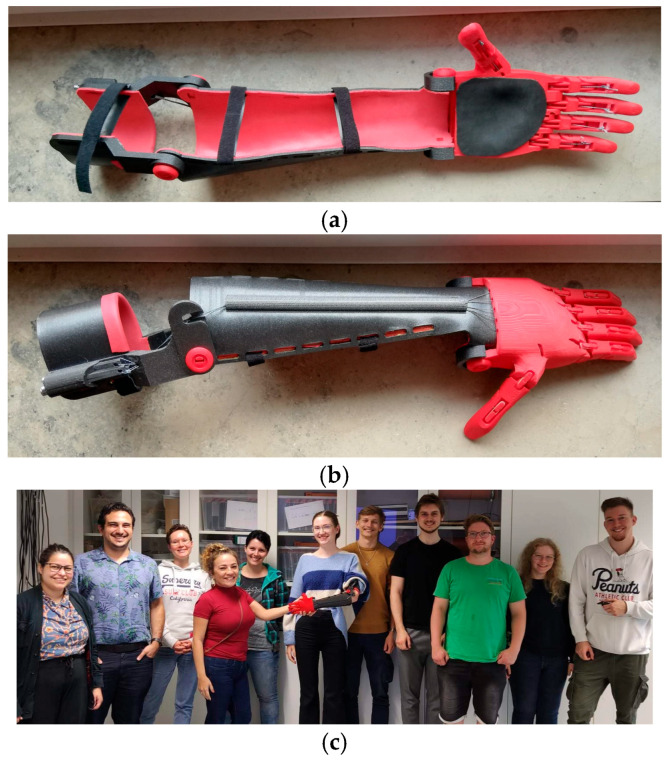
Practical module to teach the manufacturing of the Prosthesis Unlimbited Arm v2: (**a**) inferior view with padding and straps, (**b**) top view of the assembled prosthesis. Student team presenting the final result of this module (**c**).

**Figure 4 biomimetics-10-00391-f004:**
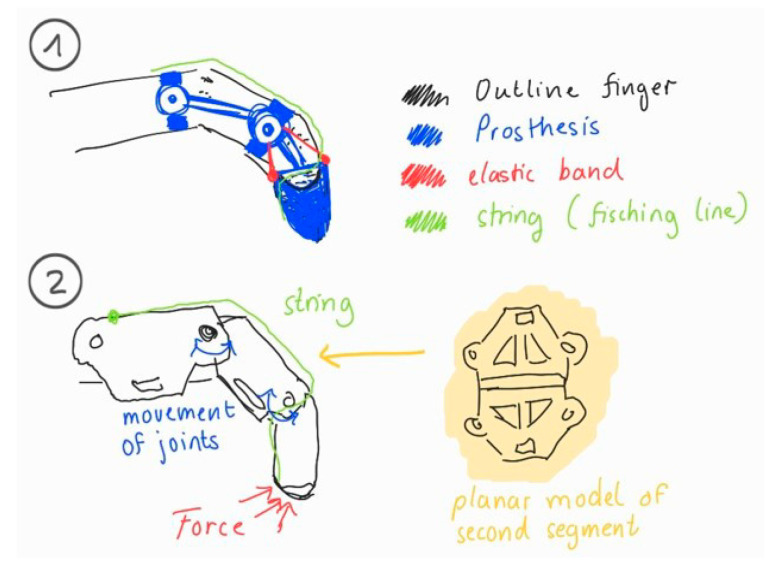
Ideation sketches for finger prosthesis: (**1**) Elastic and string-based actuation; (**2**) joint motion and internal force flow.

**Figure 5 biomimetics-10-00391-f005:**
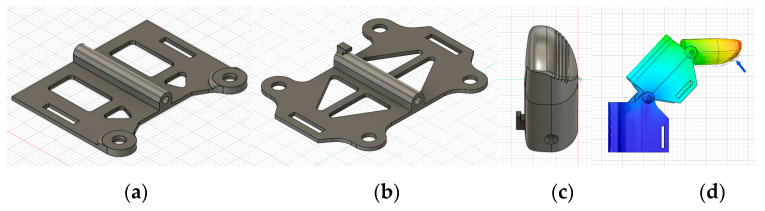
Digital modeling and simulation of the prosthetic finger. (**a**) Distal phalanx CAD model; (**b**) middle phalanx CAD model; (**c**) fingertip CAD view; and (**d**) FEM analysis of stress distribution.

**Figure 6 biomimetics-10-00391-f006:**
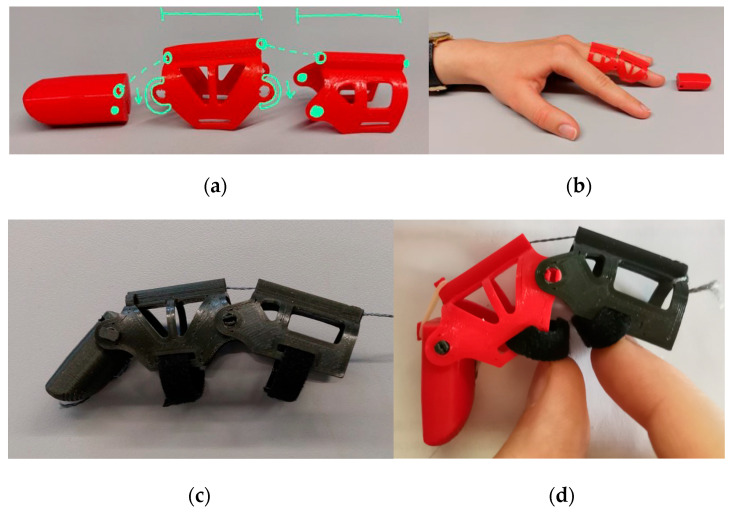
Prototyping stages of the finger prosthesis. (**a**) First printed parts of the prototype with highlighted areas for improvement. (**b**) Initial fitting test of the prosthesis on the student’s hand. (**c**) Complete prototype with first design modifications. (**d**) Improved version of the prosthesis during articulation test.

**Figure 7 biomimetics-10-00391-f007:**
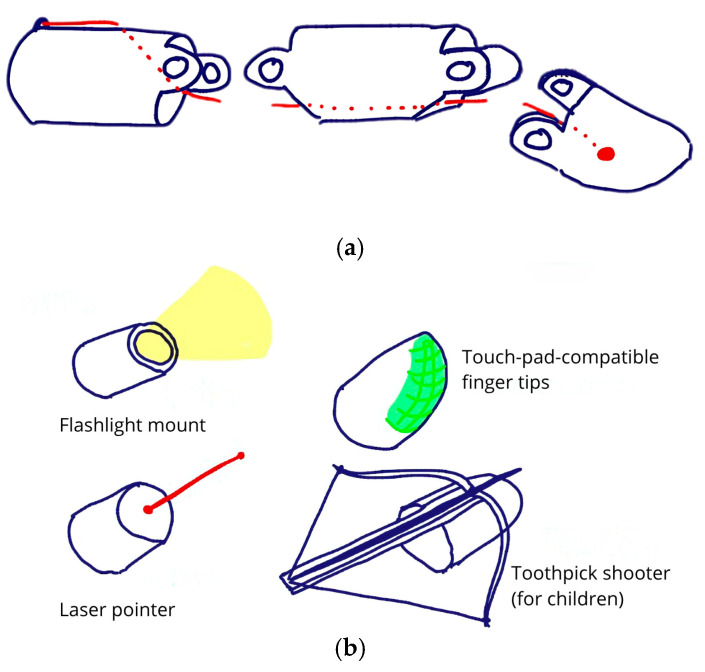
Sketches developed during the ideation phase of a finger prosthesis. (**a**) Modular prosthetic segments showing tendon routing and plug–twist attachment mechanism. (**b**) Conceptual fingertip attachments for playful and functional use, including flashlight, laser pointer, touchpad stylus, and toothpick shooter.

**Figure 8 biomimetics-10-00391-f008:**
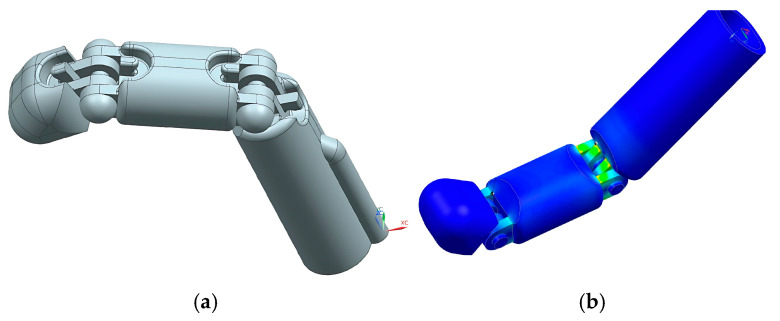
Digital model and FEM analysis of a finger prosthesis. (**a**) Final CAD model of the modular finger prosthesis. (**b**) FEM simulation highlighting stress concentrations in the joint bolts under 10 N load.

**Figure 9 biomimetics-10-00391-f009:**
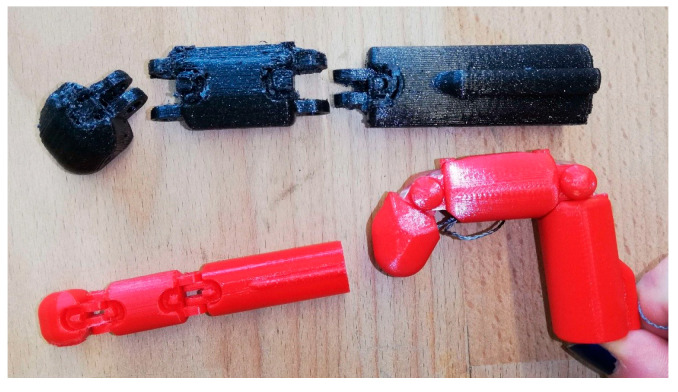
Fabrication stages of the Handy Solutions prosthesis. From left to right: first prototype (small red), second prototype (black), and final prototype (large red) with optimized joints and articulation.

**Figure 10 biomimetics-10-00391-f010:**
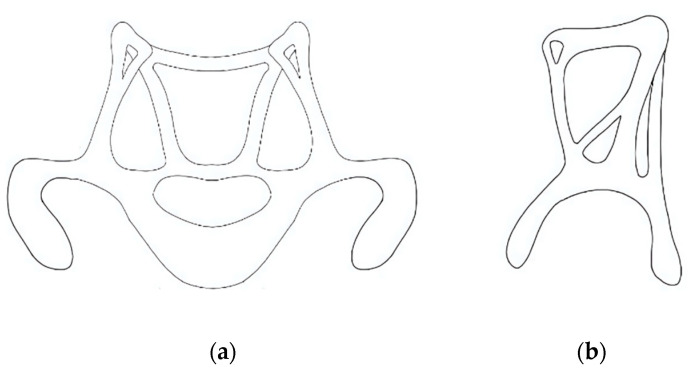
Cervical orthosis concept development based on initial hand-drawn sketches by students, showing (**a**) frontal and (**b**) side views using anatomical and geometric considerations. Images reflect authentic student outputs and were not post-processed to preserve their educational value.

**Figure 11 biomimetics-10-00391-f011:**
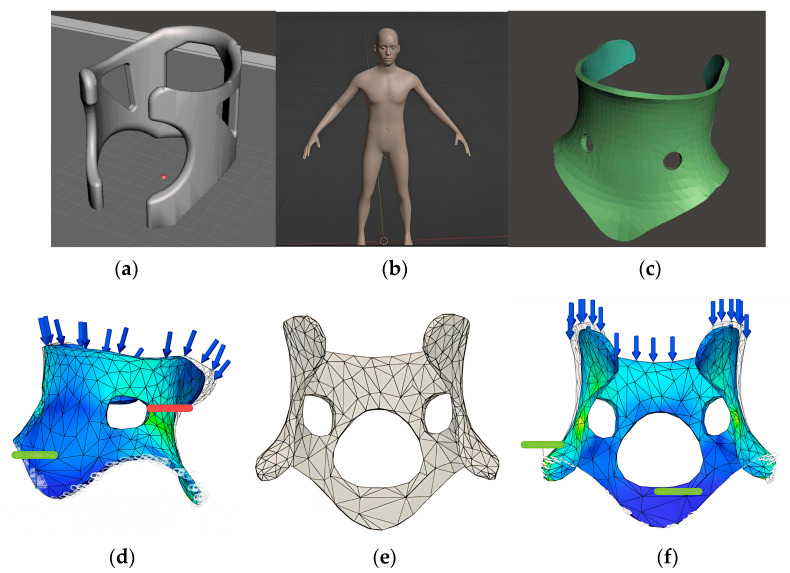
Cervical orthosis 3D modeling and simulation steps, developed by undergraduate students during the course. (**a**) Initial 3D model created with primitive geometry. (**b**) Digital human model generated in MakeHuman. (**c**) Anatomically adapted design modeled in Blender. (**d**) FEM analysis highlighting low-stress regions. (**e**) Optimized brace model with material reduction. (**f**) Final FEM simulation with topological refinement.

**Figure 12 biomimetics-10-00391-f012:**
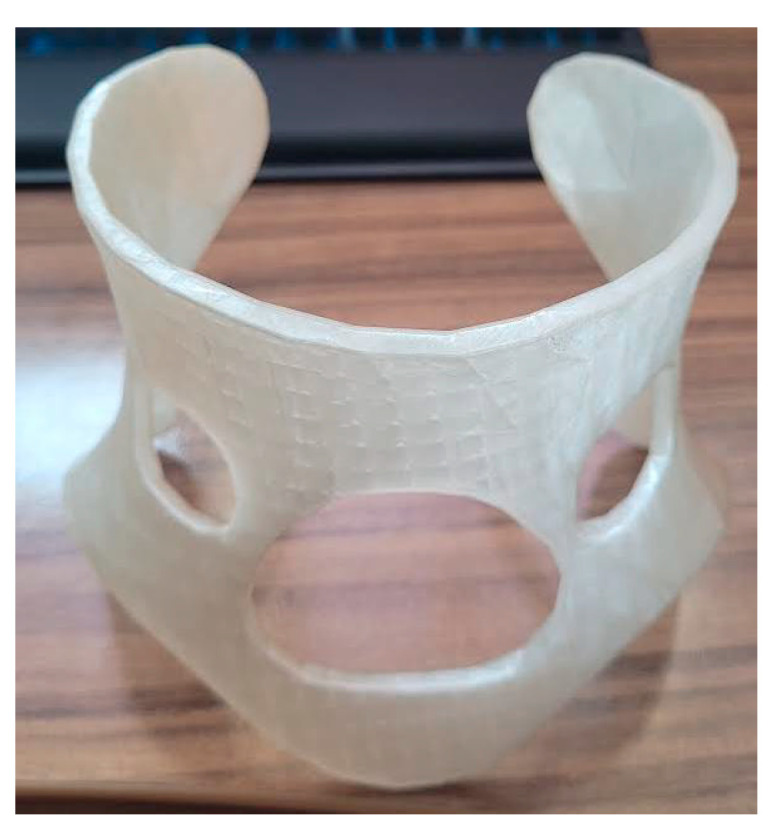
Final prototype of the cervical orthosis printed in PLA material using a Bambu Lab X1 Carbon 3D printer.

**Figure 13 biomimetics-10-00391-f013:**
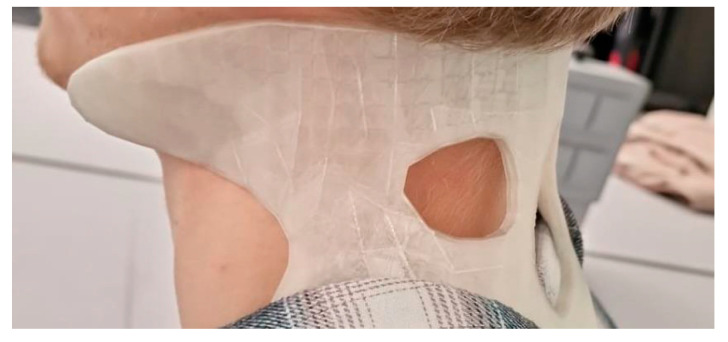
Cervical orthosis worn by a student during the testing phase.

**Figure 14 biomimetics-10-00391-f014:**
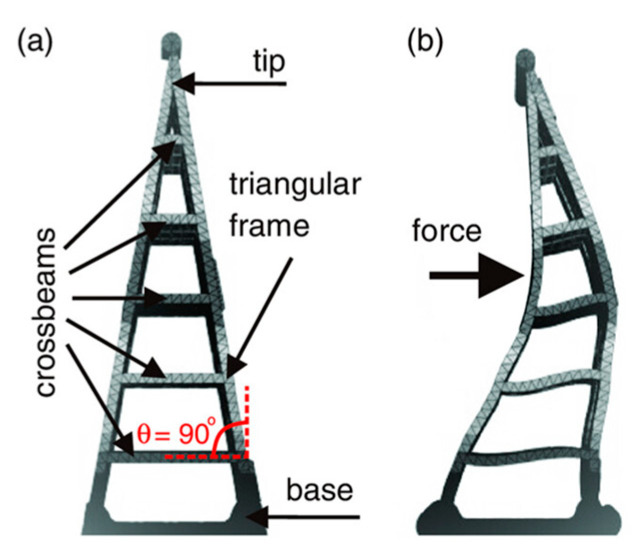
The Fin Ray effect. (**a**) Structure in neutral position with a triangular frame and crossbeams. (**b**) Deformation under applied force, bending toward the force and enabling adaptive grip.

**Figure 15 biomimetics-10-00391-f015:**
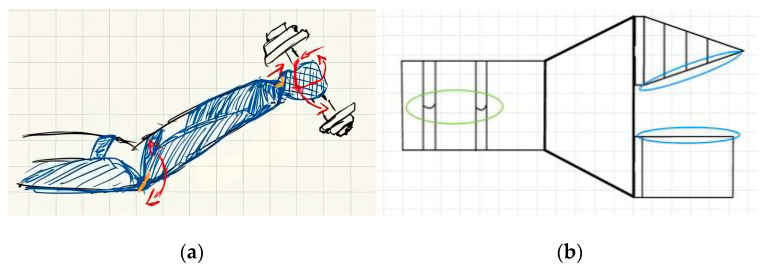
Initial design sketches for a prosthesis using the Fin Ray effect, developed by students during the ideation phase. (**a**) Early concept for a sport-specific prosthesis with embedded tools. (**b**) Modular finger configuration using Fin Ray geometry. Sketches are presented in their original form to preserve the educational and conceptual process.

**Figure 16 biomimetics-10-00391-f016:**
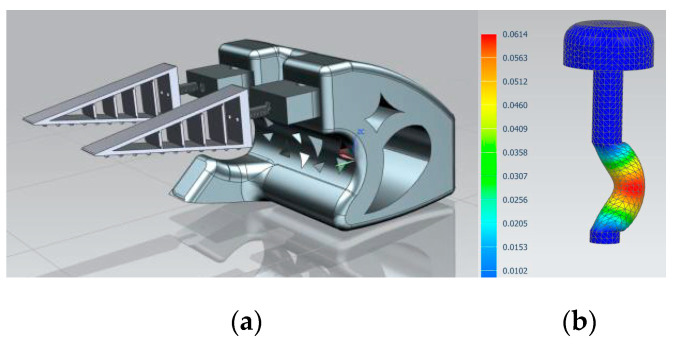
Three-dimensional modeling and FEM analysis prosthesis using the Fin Ray effect. (**a**) Final CAD model of the prosthetic hand with Fin Ray fingers. (**b**) FEM simulation showing stress distribution in the connecting pin under 100 N load.

**Table 1 biomimetics-10-00391-t001:** Summary of software used throughout the course, categorized by application.

Category	Software	Purpose
CAD	TinkerCAD (v.2023)	Basic parametric 3D modeling
	Fusion 360 (v.2023)	Parametric design and mechanical assemblies
	Siemens NX (v.2306)	High-level CAD modeling and FEM preparation
	Blender (v.3.6 LTS)	Organic modeling and sculpting for anatomical fit
	Meshmixer (v.3.5.474)	Mesh editing and adjustments of anatomical parts
	MakeHuman (v.1.2.0)	Generation of human body models
	3D Slicer (v.5.2.2)	Anatomical biomodels from medical imaging
	OpenSCAD (v.2021.01)	Parametric modeling using code-based scripts
CAE	Fusion 360 Simulation (v.2023)	Structural and mechanical analysis of components
	ANSYS (v.2023 R2)	FEM simulations for validation
	Siemens NX (v.2306)	FEA simulations for advanced validation
CAM	Ultimaker Cura (v.5.4)	Slicing and print setup for 3D printers
	PrusaSlicer (v.2.6.0)	Alternative slicing tool for 3D printers

**Table 2 biomimetics-10-00391-t002:** Summary of 3D printers used throughout the course, categorized by application.

Printer	Manufacturer (Country)	Build Volume (mm)
Prusa MK3S	Prusa Research (Czech Republic)	250 × 210 × 210
Prusa MK3		250 × 210 × 210
Prusa Mini		180 × 180 × 180
Anycubic Chiron	Anycubic (China)	450 × 400 × 380
Bambu Lab X1 Carbon	Bambu Lab (China)	256 × 256 × 256

**Table 3 biomimetics-10-00391-t003:** Eight-step PBL pedagogical model developed with biomimetics students at Westfälische Hochschule (Germany).

Step	Phase	Description
1	Problem definition	Problem identification and user needs analysis
2	Biological research	Research and selection of biological analogies
3	Ideation	Ideation of the assistive device using semantic panels and design tools
4	Decision-making	Decision matrix for material selection and concept feasibility
5	Modeling and simulation	3D modeling and mechanical simulation using FEM
6	Fabrication	Digital fabrication using FFF 3D printing
7	Testing and evaluation	User scenario simulation using AI-based contextual analysis
8	Presentation and reflection	Final presentation and peer reflection

This model was adapted from Stern et al. [53].

**Table 4 biomimetics-10-00391-t004:** Summary of student groups, project type, team size, and biomimetic inspiration.

Project Title	Type	Members	Biomimetic Inspiration
Finger Prosthesis for Musician	Prosthesis	3	Bird phalanges
Handy Solutions	Orthosis	2	Tendon–muscle antagonism
Cervical Brace	Orthosis	4	Stingray form
Fin Ray-Inspired Hand	Prosthesis	4	Fin Ray effect (fish fins)

Group 1: H.H., L.S., and A.T.; Group 2: J.K. and F.W.; Group 3: H.H., D.M., T.T., and L.M.; and Group 4: S.K., A.Ö., B.U., and J.W.

## Data Availability

The data presented in this study are available upon request from the corresponding authors. The data are not publicly available due to confidentiality.

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
