# Peer review of "Teaching Bioinspired Design for Assistive Technologies Using Additive Manufacturing: A Collaborative Experience"

_biomimetics, 2025, doi:10.3390/biomimetics10060391_

Round 1
Reviewer 1 Report
Comments and Suggestions for Authors
The presented article presents an innovative approach to the field of education of future engineers, applying bioinspired design and the use of additive technologies. The authors describe the results of an interdisciplinary course focused on the application of biomimetics and additive manufacturing (3D printing) in the context of designing and manufacturing low-cost prosthetic and orthotic devices. The course was conducted in cooperation between a German and a Brazilian university in a bilingual format and was based on the principles of project-based learning (PBL) and Design Thinking.
The article is well structured and professionally written. The title and abstract describe the focus and main objectives of the work. The authors have demonstrated a high degree of originality - whether in the teaching methodology itself, in linking theory with practice, or in emphasizing the social and ethical dimension of technical innovation. At the same time, they describe the infrastructure used, software and hardware equipment, as well as pedagogical strategies.
The idea of ​​the course was to connect multiple disciplines (engineering, design, biology and social sciences) and provide a basis for students to create real, personalized solutions. The benefit is the individual steps of the module, where each team solved a specific case, was inspired by natural patterns (e.g. stingrays, bird phalanges, Fin Ray effect) and went through the entire cycle from empathic understanding to design, simulation, 3D printing and prototype testing.
The results confirm that students not only acquired technical skills in the areas of CAD, FEM simulations and additive manufacturing, but also acquired critical thinking, teamwork skills and sensitivity to the needs of disadvantaged groups. An important aspect of the article is also the emphasis on open technologies (open-source solutions) and low-cost approaches, which are important for expanding the availability of prosthetic and orthopedic devices, especially in countries with limited resources.
From a methodological point of view, the authors consistently describe the course of instruction, student evaluation criteria, and qualitative outputs that were obtained through presentations, peer-review, and group reflections. They also state the limitations of the study – the short duration of the course, the absence of testing with real users, and some technical obstacles in 3D printing.
I wish the authors continued success in organizing the course "3D Prosthetics and Orthotics" and perhaps they could also create a similar course for researchers or. teachers at universities - especially in terms of teamwork and critical thinking.
I recommend only slight adjustments regarding shortening repetitive passages (especially in project descriptions) and expanding suggestions for involving real users in future iterations of the course.
Author Response
Response to Reviewer 1
We would like to thank Reviewer 1 for the thorough and constructive evaluation of our manuscript. We greatly appreciate the encouraging comments regarding the originality, structure, and pedagogical innovation presented in the article. Below, we address the specific suggestions and confirm the revisions made:
Comment 1:
“I recommend only slight adjustments regarding shortening repetitive passages (especially in project descriptions).”
Response:
Thank you for this observation. We carefully revised Section 3.3 and its subsections (3.3.1 to 3.3.4) to eliminate redundant wording and streamline the narrative, while preserving all relevant technical and pedagogical details. This has improved clarity and readability without affecting the richness of the students' design process.
Comment 2:
“…and expanding suggestions for involving real users in future iterations of the course.”
Response:
We fully agree with this suggestion. A paragraph was added at the end of Section 3.3 to emphasize the potential role of real users in future course iterations, particularly through co-design workshops and user testing. Additionally, this recommendation was integrated into the final paragraphs of both the Discussion and Conclusion sections, reinforcing the importance of early stakeholder involvement and the translational value of student projects.
Excerpt from Section 3.3:
“In future iterations of the course, involving real users in selected stages (such as testing or co-design workshops) could enhance the translational potential of the outcomes and foster deeper empathy.”
Excerpt from Conclusion:
“…future iterations of the course should incorporate extended timelines, early involvement of real users, and structured follow-up mechanisms—such as clinical testing, co-design workshops, and mentorship programs…”
We sincerely thank the reviewer for the insightful feedback and encouraging evaluation. The suggestions helped improve the overall structure and impact of the manuscript. All changes have been implemented with care, and we hope the revised version meets the journal’s expectations.
Reviewer 2 Report
Comments and Suggestions for Authors
As it follows from the reviewed text title ("Teaching Bioinspired Design for Assistive Technologies Using") – the draft scientific article is mainly devoted to the issues of the correct educational process construction. It becomes clear from the content analysis that the text is actually a form of a report on the procedure and the results of implementation of an additional educational program within the framework of an international scientific-educational consortium functioning. According to the reviewer’s opinion, the work is characterized by high relevance and practical significance in the field of methodology of highly motivating educational process organizing.
But it is wellknown that the «Biomimetics» is an open access journal regarding biomimicry and bionics. It is dedicated to research that relates to the most basic aspects of living organisms and the transfer of their properties to human applications.
The Journal invites submissions on a wide range of topics:
1) Biomimetic mechanisms and design.
2) Biomimetic robotics.
3) Biofabrication and characterization.
4) Biomimetic and bioinspired chemistry.
5) Bioinspired sensing.
6) Nanotribology, nanomechanics, micro/nanoscale studies on biological systems.
7) Biomimetics inspired by animal and plant biomechanics.
8) Synthetic and biohybrid systems.
9) Self-organization and cooperative behavior.
10) Biomimetic aspects of tissue engineering.
11) Bioinspired, biomedical and biomolecular materials.
It is not possible to relate the content of the presented text in the "as is" state to any of these topics.
Thus, the reviewer does not recommend the submitted text for publication in the «Biomimetics» Journal. According to the reviewer’s opinion, if the authors want to publish their results in this particular Journal, it is necessary to almost completely exclude from the text all descriptions of the highly effective educational technologies they use, focusing on the analysis and detailed study of models of biological prototypes, methods of their technological adaptation and methods of production and testing of the products’ properties.
Author Response
Response to Reviewer 2
We thank the reviewer for the thoughtful feedback and for raising important points regarding thematic alignment with the scope of the journal Biomimetics. We would like to respectfully offer some clarifications and highlight how the manuscript fits within the journal’s outlined topics.
Although the article is situated within an educational framework, its core contributions lie in the application of biomimetic principles to the design and fabrication of assistive technologies. Each student project developed during the course was inspired by biological systems—such as the Fin Ray effect, bird phalanges, and stingray morphology—and resulted in a functional prototype combining CAD design, FEM simulation, and additive manufacturing workflows. These projects align closely with several of the journal’s themes, including:
- Biomimetic mechanisms and design (Topic 1);
- Biomimetics inspired by animal biomechanics (Topic 7);
- Biofabrication and characterization (Topic 3);
- Bioinspired, biomedical and biomolecular materials (Topic 11).
The manuscript reports the development, simulation, and fabrication of bioinspired devices, focusing on the translation of natural models into engineered solutions. The course served not merely as an instructional setting, but as a research-based environment where students applied biomimetic analogies to solve real-world problems in assistive technology.
Moreover, the open-source and low-cost approach adopted in the course contributes to advancing decentralized, personalized innovation in biomedical device design, particularly relevant in low-resource contexts. These contributions go beyond pedagogical interest and represent actionable outcomes grounded in biomimetic research and fabrication methods.
It is also important to emphasize that the pedagogical strategy itself is relevant to the advancement of the field of biomimetics. There is a growing recognition that training future engineers and designers in biomimetic thinking, through applied, interdisciplinary, and international learning experiences, is essential for the consolidation and dissemination of this area of knowledge. The course described in the manuscript provides a replicable model for achieving this goal.
In response to this valuable feedback, we have also revised parts of the Abstract, Discussion, and Conclusion sections to make the technological and biomimetic aspects of the study more prominent, while preserving only the essential context needed to understand the development pathway of each prototype.
We hope these clarifications demonstrate that the manuscript meaningfully contributes to the field of biomimetics and remains within the journal’s intended scope. We remain available for any further clarification or revision the editorial board may require.
Round 2
Reviewer 2 Report
Comments and Suggestions for Authors
The authors presented an improved version of the article. Considering that the article is educational and scientific in nature and is devoted to biomimetic principles in “3D Prosthetics and Orthotics technology. Recommended for publication in the present form